# Preeclampsia and Cerebral Palsy in Offspring

**DOI:** 10.3390/children9030385

**Published:** 2022-03-09

**Authors:** Miltiadis Badagionis, Theodoros N. Sergentanis, Panagiota Pervanidou, Emmanouil Kalampokas, Nikolaos Vlahos, Makarios Eleftheriades

**Affiliations:** 1Unit of Surgical Oncology, Second Surgical Department, “Korgialeneio—Benakeio”, Red Cross Athens General Hospital, 115 26 Athens, Greece; miltosbadag1@icloud.com; 2Second Department of Obstetrics and Gynaecology, Aretaieio Hospital, National and Kapodistrian University of Athens, 115 28 Athens, Greece; m.kalampokas@gmail.com (E.K.); nfvlahos@gmail.com (N.V.); 3Department of Public Health Policy, University of West Attica (UNIWA), 122 43 Athens, Greece; tsergentanis@yahoo.gr; 4Unit of Developmental and Behavioral Pediatrics, First Department of Pediatrics, School of Medicine, National and Kapodistrian University of Athens, 115 27 Athens, Greece; nenyperva@gmail.com

**Keywords:** preeclampsia, cerebral palsy, offspring, prematurity, gestational age

## Abstract

The aim of this systematic review and meta-analysis is to examine the association between exposure to preeclampsia during pregnancy and the occurrence of cerebral palsy in offspring. For this reason, the authors searched PubMed/Medline, EMBASE, and Google Scholar databases (end-of-search: 22 November 2021) and identified the most relevant studies. Then, a meta-analysis of all the eligible studies was performed. Subgroup and meta-regression analyses by study design, degree of adjustment, and geographical region were also conducted. A total of 10 studies were finally included, and no statistical significance was noted in the association between preeclampsia and cerebral palsy (pooled OR = 1.16, 95% CI: 0.77–1.74). The subgroup of studies that provided adjusted odds ratios for any variable except for gestational age showed a statistically significant association (pooled OR = 1.62, 95% CI: 1.36–1.93), whereas the association dissipated in studies also adjusting for gestational age (pooled OR = 1.63, 95% CI: 0.48–5.50). In conclusion, it seems that preeclampsia is not associated with cerebral palsy independently of gestational age; however, further research is needed to shed light on this topic.

## 1. Introduction

Preeclampsia (PE) is a multisystemic pregnancy disorder and can affect up to 8% of all women during gestation in developed countries. It is characterized by hypertension and proteinuria or hypertension and significant end-organ dysfunction with or without proteinuria. It can range from mild to severe form. Additionally, it can be developed early or late in pregnancy, but sometimes even after delivery during puerperium [1,2]. However, it has been proposed that early and late PE are not two different clinical entities, but rather a spectrum of the same disorder, the degree of which is reflected in both the gestational age at the time of delivery and the severity of the disease based on clinical and laboratory findings [3].

The diagnosis of preeclampsia is based on increased blood pressure (systolic blood pressure > 140 mmHg or diastolic > 90 mmHg) that can be verified in two different measurements with a 4 to 6 h interval accompanied with proteinuria ≥ 300 mg/24 h or 1+ on urine dipstick or protein/creatinine ratio ≥ 30 mg/mmol [4,5,6]. However, the diagnosis of preeclampsia can still be made in the absence of proteinuria if hypertension is accompanied by specific signs or symptoms indicating significant end-organ dysfunction, such as central nervous involvement (severe headache, photopsia, and scotomata), impaired hepatic function, renal insufficiency, thrombocytopenia, and pulmonary edema [7,8,9].

Risk factors for preeclampsia include a previous history of preeclampsia, chronic hypertension, autoimmune diseases (systematic lupus erythematosus and antiphospholipid syndrome), chronic kidney disease, gestational diabetes, nulliparity, twin pregnancy, a family history of preeclampsia, maternal overweight or obesity, and previous history of placental insufficiency, such as fetal growth restriction (FGR), stillbirth, and placental abruption [5,7,10]. The pathophysiology of preeclampsia is associated with environmental, maternal, and placental factors. Defects in the development of placental vasculature early in pregnancy, such as abnormal remodeling of spiral arteries and defective trophoblast differentiation, may result in placental hypoperfusion, hypoxia, and ischemia. The consequence of the abnormal placentation is reduced production and expression of angiogenic parameters, such as vascular endothelial growth factor (VEGF) and placental growth factor (PlGF), and increased placental expression and secretion of antiangiogenic factors, such as soluble fms-like tyrosine kinase 1 (sFlt-1), into maternal circulation, causing maternal systematic endothelial dysfunction. The outcome of the altered endothelial function is maternal hypertension and further clinical signs of the disease, such as proteinuria and renal insufficiency, platelet activation and hemolysis, ischemia, necrosis and dysfunction, central nervous system manifestations and visual disturbances, and cardiac and pulmonary dysfunction. However, the trigger for abnormal placental development and the subsequent cascade of events remains unknown [5,7,11,12,13,14].

Nowadays, screening tests are used early in gestation to identify high-risk pregnancies for developing preeclampsia. Based on historical risk factors (maternal and pregnancy characteristics), only approximately 30% of women who will develop preeclampsia can be predicted [15,16,17,18]. Particularly, in the context of personalized prediction, emphasis is given to maternal characteristics, medical and obstetric history, and biochemical and biophysical measurements [9,19,20,21,22]. In terms of prevention, there is strong evidence that, in pregnancies at high risk of PE, the administration of aspirin (150 mg/day) reduces the rate of early PE (<32 weeks) by about 90% and preterm PE by 60% [9,19,20,21]. Furthermore, lifestyle changes such as weight loss in obese women, avoiding excessive gestational weight gain, and multifetal pregnancies following assisted reproductive technology (ART) for infertility treatment have been associated with a significant reduction in the risk of developing preeclampsia. PE has been associated with increased perinatal mortality and morbidity, as it is a major risk factor for developing serious complications of eclampsia and hemolysis, elevated liver enzymes and low platelets (HELLP) syndrome, prematurity, and long-term maternal disease, such as cardiovascular disease, kidney disease and type 2 diabetes [9,23]. The aim in management of this disease is to maintain blood pressure at acceptable levels below 150 mmHg for the systolic blood pressure and 80–100 mmHg for the diastolic blood pressure. In case blood pressure deviates from the normal levels for more than 15 min and systolic blood pressure exceeds 160 mmHg and/or diastolic blood pressure is increased above 110 mmHg, immediate treatment with intravenous administration of labetalol or hydralazine or oral nifedipine is the appropriate solution. However, if there are signs of seizures or fetal distress, then delivery is required [5,6,9].

Cerebral palsy (CP) is the result of abnormalities of the developing fetal or infantile brain [24]. This is the most common motor disorder in children, and its prevalence is estimated to vary from 1.5% to 3% [25]. As far as its risk factors are concerned, preterm birth is a major one, and if delivery takes place before 28 weeks, the risk of cerebral palsy is significantly higher; other complications such as inflammation during pregnancy, perinatal hypoxic-ischemic injury, and hypertensive disorders leading to preterm birth have also been correlated with cerebral palsy [25,26].

A diagnosis of cerebral palsy can be made clinically on the basis of perinatal history, neurological examination, imaging testing, and laboratory assessment. Early detection of cerebral palsy before 5 months of age can be evaluated with magnetic resonance imaging (MRI) of the infant, the Prechtl Qualitative Assessment of General Movements (GMA), and the Hammersmith Infant Neurological Examination (HINE). Infants with an identified risk factor related to cerebral palsy must be considered “high risk” and must be closely monitored for administration of every useful early treatment. In most cases, the diagnosis can be established between 12 and 24 months of life. Different subtypes depending on the affected brain area, motor disability type, and the level of functionality have been described [24,27]. In terms of prevention, measures include the prediction and prevention of preterm labor, the administration of magnesium sulfate for neuroprotection in preterm pregnancies, delayed umbilical cord clamping, the maintenance of sufficient ventilation, cerebral perfusion, metabolic status, and therapeutic hypothermia [26,27]. As far as management is concerned, the earlier the diagnosis of cerebral palsy, the better the outcome for newborns’ motor and spasticity recovery, aiming to maximize daily functional activities’ independence and decrease the extent of disability [27,28].

Hypertensive disorders of pregnancy and particular preeclampsia have been associated with an increased risk of neurodevelopmental disorders and cerebral palsy in childhood. Pregnancy comorbidities such as preterm birth, gestational diabetes, and fetal growth restriction seem to further increase the abovementioned risk. Furthermore, the severity of hypertensive disorder and gestational age at the onset of the disease may contribute to developmental impairment and neurological sequelae. However, even preeclampsia at term has a lasting effect on neurodevelopment of the offspring. Data regarding the association between preeclampsia and infantile cerebral palsy are conflicting, though there are studies supporting that the overall likelihood of cerebral palsy is reduced in the offspring of preeclamptic mothers irrespective of magnesium administration, and this risk is mainly limited in cases complicated by preterm birth [26,29,30,31,32]. The mechanisms of this controversial association remain to be elucidated; nevertheless, placental insufficiency and the associated oxygen and nutrients deprived in utero environment have been proposed as a possible etiology [33]. The aim of the present study is to conduct a systematic review and meta-analysis of all the available data that have assessed the association between preeclampsia and the risk of cerebral palsy in offspring.

## 2. Materials and Methods

### 2.1. Search Strategy and Eligibility of Studies

The present systematic review and meta-analysis was performed according to the Preferred Reporting Items of Systematic Reviews and Meta-analyses (PRISMA) guidelines [34], and the completed PRISMA statement is available on Appendix A. Additionally, the Systematic Review registration statement is available on PROSPERO (ID:306467). The study protocol was discussed, and there was a broad consensus among all authors. A careful search was performed in PubMed/Medline and EMBASE (end-of-search: 22 November 2021) based on the following algorithm: (preeclampsia OR pre-eclampsia OR eclampsia OR pre-eclamptic OR preeclamptic) AND (“cerebral palsy”). No restriction was set regarding the publication language. At the same time, we performed a search in all the studies used for analysis for some data within the subject of this review in a “snowball” procedure. Additionally, we performed a search in Google Scholar using the keywords “pre-eclampsia” and “cerebral palsy”, and we examined the first 300 hits, aiming to find additional relevant studies.

The eligible studies for our analysis were randomized controlled trials, case–control and cohort studies, but not case reports or case series. Search and selection of the studies was made by two reviewers (MB, ME) who worked independently, and any disagreement was resolved with the broad consent of all authors. A very crucial fact in our selection criteria was that the follow-up should be started immediately after birth, as the diagnosis of cerebral palsy most likely occurs between 12 and 24 months of life [24,27]. Additionally, among the studies that we identified for analysis, we excluded some because of population overlapping, and we included the larger ones.

### 2.2. Data Collection and Effect Estimates

Data extraction was carried out based on the general background of every study (first author’s name and year of publication), characteristics (study design and period of interest, geographical region), and follow-up period of the newborns. At the same time, data extraction was carried out for cohort size and cases of cerebral palsy (for cohort studies), number of cases and controls (for case–control studies), features of mothers with preeclampsia and infants with cerebral palsy, and the main results of every study, including the factors adjusted for in the multivariate analyses. As in the selection of the studies, the two reviewers (MB, ME) extracted data autonomously, and then the writing team, after consultation, concluded on what was appropriate and useful for the analysis. Odds ratios, along with their 95% confidence intervals (95% CI), as well as relevant data for calculation, were abstracted from each study; whenever possible, adjusted effect estimates were preferred over unadjusted ones.

### 2.3. Synthesis of Results

Random effects (DerSimonian–Laird) models were implemented for the estimation of pooled odds ratios and 95% CIs. As far as heterogeneity between studies is concerned, this was evaluated by Q-test and I^2^ [35]. Subgroup analyses were performed by degree of adjustment (unadjusted; adjusted for variables except for gestational age; adjustment for variables including gestational age), geographical region (Europe, USA), or study design (case–control or cohort) in focus in order to find out possible reflect to the results. We used STATA/SE version 13 (Stata Corp, College Station, TX, USA) for all our analyses.

### 2.4. Assessment of Quality and Publication Bias

Authors examined the quality of the studies included in the analysis via Newcastle–Ottawa Quality scale, and study quality was considered “low” when the Newcastle–Ottawa score measured between 1 and 3, “intermediate” when the score was between 4 and 6, and “high” when the score ranged between 7 and 9. The two researchers (MB, ME) again worked independently and evaluated the studies regarding the quality based on this scale and recorded their results. Additionally, it was very important for the follow-up period in every study to be checked, and in advance, the authors set the restriction of immediate start after birth, based on what was already known for cerebral palsy. Then, consensus followed.

We evaluated the possible existence of publication bias using Egger’s formal statistical test [36] and visual inspection of the funnel plot. The level of statistical significance for this test was set at *p* < 0.1. For this test, we again used the STATA/SE version 13 (Stata Corp, College Station, TX, USA).

## 3. Results

### 3.1. Features of Eligible Studies

During the search in the literature, we identified 559 studies that met our eligibility criteria (152 from PubMed/Medline, 300 from Google Scholar, and 107 from EMBASE), and all the steps of our selection process are available in Appendix A. When all the studies were gathered, we excluded 289 studies based on the title or the abstract as irrelevant or as duplicate. Then, the 270 studies remained for further full-text evaluation. Out of this number of studies, three were excluded due to overlapping in the populations that were examined, one because it also included cases of pregnancy-induced hypertension without separation of the cases of preeclampsia, and one because of a lack of provided odds ratio for the association that this analysis examines (Appendix A).

Finally, 10 studies were considered eligible for this systematic review. Two studies followed the case–control design [37,38], and eight followed the cohort style [39,40,41,42,43,44,45,46]. All the characteristics of the studies included are presented in Appendix A. Additionally, all the studies were evaluated based on Newcastle–Ottawa Scale, and the results of this rating are presented in Appendix A. Quality scores ranged between 6 and 9.

### 3.2. Synthesis of Studies and Meta-Analysis

All the 10 studies were included in the overall analysis. All the results of the general analysis and the subgroup analyses are presented in Appendix A. In the overall analysis, there was no statistical significance regarding the association between preeclampsia and cerebral palsy (pooled OR = 1.16, 95% CI: 0.77–1.74).

Three studies [40,44,45] provided adjusted odds ratios for several variables except for gestational age; three studies [38,39,43] presented a multivariate analysis, adjusting also for gestational age; and four remaining studies [37,41,42,46] provided unadjusted odds ratios. A significant association arose only in the subgroup of studies that performed a multivariate analysis, adjusting for variables other than gestational age (pooled OR = 1.62, 95% CI: 1.36–1.93). In the subgroup of studies that performed a multivariate analysis including gestational age, the pooled was equal to 1.63 (95% CI: 0.48–5.50), and in the subgroup of studies that performed univariate analysis, the pooled OR was 0.65 (95% CI: 0.20–2.18). These results are presented in Figure 1 (Appendix A).

Regarding geographical region, there was no association either in studies of European populations, including in Turkey and Israel, or in USA (Europe OR = 1.22, 95% CI: 0.68–2.20 and USA OR = 1.06, 95% CI: 0.53–2.13). The analysis in the subgroup of cohort studies did not find any association either (Cohort OR = 0.93, 95% CI: 0.61–1.41). However, a significant association was noted in the subgroup of case–control studies (pooled OR = 5.00, 95% CI: 2.17–11.50), but in this group, the number of studies was only two. All these data are presented in Figure 2 (Appendix A) and Figure 3 (Appendix A). No statistically significant publication bias (*p* = 0.456) was identified, according to Egger’s statistical test and the funnel plot is also available on Figure 4 (Appendix A).

## 4. Discussion

This systematic review and meta-analysis focused on preeclampsia and the possible independent association with cerebral palsy in the offspring of such mothers after comprising the data from 10 studies, which were eligible according to the authors’ criteria. The analysis that took place pointed to an overall lack of an independent association between preeclampsia and cerebral palsy, especially after adjusting for gestational age.

The two case–control studies that we included were based on specific population groups in Estonia and Turkey and tried to find out the most related factors to the development of cerebral palsy. Stelmach et al. compared cases diagnosed with mild to severe impairment due to cerebral palsy with two controls for every case from the same population, aiming to include all cases with a mild form of cerebral palsy, and concluded that preeclampsia could be an important parameter for the onset of this disease (OR = 8.54, 95% CI: 1.84–39.62), especially since this result was adjusted for several factors including gestational age [38]. On the other hand, Ozturk et al. made a univariate analysis by including all children with cerebral palsy, regardless of its severity, in comparison with healthy controls arising from various guaranteed sources in a specific region of Turkey. This study also highlighted a greater number of pregnancies complicated with preeclampsia in the group of cases, rather than in the group of controls (OR = 4.00, 95% CI: 1.48–10.78) [37]. In both studies, the results were statistically significant, but we must consider that there were some design issues.

Eight cohort studies were considered compatible with our criteria for analysis. Withagen et al. included three groups of infants, one coming from mothers that underwent treatment for a severe form of preeclampsia before 32 weeks of pregnancy and two others related to normal pregnancies with infants born either on admission time of preeclamptic mothers or at the same gestational age as the infants of these mothers. By comparing these groups without adjusting for any factor, the study showed that there is no statistically significant result of possible association between preeclampsia and cerebral palsy (OR = 0.44, 95% CI: 0.16–1.22) [46]. Two other studies that followed the univariate analysis model examined the prevalence of cerebral palsy among infants born very preterm [41] and infants with very low birth weight [42], trying to find possible association with many perinatal risk factors. Both studies concluded that in these special categories of newborns, preeclampsia may act protectively for the development of cerebral palsy, and the results were enhanced by statistical significance.

Mann et al. and Tronnes et al. tried to find out the independent association and included large population groups in their analyses. The two studies managed to identify that preeclampsia, especially the early onset form, is strongly related to the development of cerebral palsy later, after adjusting for several factors but not for gestational age. Particularly, the first study was based on secured records from the South Carolina Medicaid program and demonstrated that the earlier the diagnosis of preeclampsia is set, the higher is the risk for cerebral palsy (OR = 1.90, 95% CI: 1.20–3.00) [40], and the second one also showed that many perinatal complications that occur in premature pregnancies (<32 weeks) have been associated with cerebral palsy, but especially for preeclampsia, the adjusted OR was 1.70 with 95% CI: 1.47–1.96 [45]. The same pattern was followed by a study based on a Norwegian population that performed a multivariate analysis without adjusting for gestational age. When the authors focused on term pregnancies, the study failed to conclude with statistical significance on the association examined (OR = 1.30, 95% CI: 0.94–1.80). However, in the group of preterm infants, preeclampsia seemed to have a significant effect on the possibility for cerebral palsy, but that result is certainly affected by the complications of prematurity [44].

Two studies in the group of cohort ones included gestational age as a variate in the statistical analysis. Love et al. examined all newborns from women that underwent preeclampsia in pregnancy in Aberdeen, taking advantage of records about these pregnancies in databases of this region, and showed that there was no association with the later presence of cerebral palsy (OR = 1.26, 95% CI: 0.43–3.69), even though the univariate analysis had estimated an increased existing risk of cerebral palsy [39]. In the last analysis, Strand et al. led to many conclusions depending on the level of adjustment, namely univariate analysis, small adjustment for gestational age, and small adjustment for gestational age. The study did not find any association in the group without any adjusted factor, but showed enhanced association in the group of infants that were small for gestational age and a statistically significant reduced association when the study population was adjusted for gestational age (OR = 0.73, 95% CI: 0.56–0.96) [43].

The results of our analysis are linked to the conclusions of another systematic review that examined factors responsible for the onset of cerebral palsy. Going back to 1998, Collins et al. tried for the very first time to clarify the association between preeclampsia and cerebral palsy based on studies that had taken place at that time. This failed to find association, but it was understood that preeclampsia caused preterm births and babies that were small for gestational age, which may have cerebral palsy, and this agrees with the findings of our systematic review [47]. Van Lieschout et al. in 2016 described possible factors for the development of cerebral palsy. Although this study showed that there are conflicting data for the association between chorioamnionitis and gestational age with cerebral palsy, most of them converge to a positive association. The strongest association was with low birth weight, whereas as far as preeclampsia is concerned, the study concluded that the data are limited to some primary studies, and the results failed to provide a clear answer in this field (OR = 0.91, 95% CI: 0.35–2.41), as in our study [48].

Clark et al. in 2008 described the possible association between preeclampsia and cerebral palsy, and from the studies that were taken into consideration, concluded that preeclampsia may not directly affect the risk for this disorder, and it may be the gestational age at birth that affects the results [29]. This agrees with the results of our meta-analysis in general and the results of some primary studies included that found that gestational age may be considered a mediator between preeclampsia and cerebral palsy [43]. It seems, therefore, that preeclampsia itself might not cause cerebral palsy, and there are probably mediating factors, as gestational age or babies small for gestational age are important effects in the development of this disorder [43].

Commenting in subgroup analyses, a significant association arose only in studies that adjusted for several covariates, except for gestational age; also, in the synthesis of the two eligible case–control studies, a statistically significant correlation emerged. Therefore, from a methodological point of view, the present systematic review and meta-analysis points to the need for well-designed cohort studies, adjusting for gestational age among other covariates.

In terms of limitations to the present systematic review and meta-analysis, there were no studies originating from East Asia, Africa, or South America. Another important limitation was the considerable heterogeneity of the studies included, but this was, to some extent, explained by the different approaches to the association, including different sets of covariates in the multivariate models and variable study designs. Additionally, other reasons for heterogeneity might be the different follow-up period among the studies and geographical region mentioned above.

However, among the strengths of this analysis, it the fact that we conducted an extensive search on three of the most important online databases. Then, we prepared our analysis and described our results based on the PRISMA guidelines [34]. Another important fact in our analysis was that there was no statistically significant publication bias.

## 5. Conclusions

In conclusion, the analysis of the currently available data suggests that preeclampsia does not seem to be independently associated with the odds of cerebral palsy in offspring. However, because of conflicting results, additional future cohort studies based on well-designed protocols are needed.

## Figures and Tables

**Figure 1 children-09-00385-f001:**
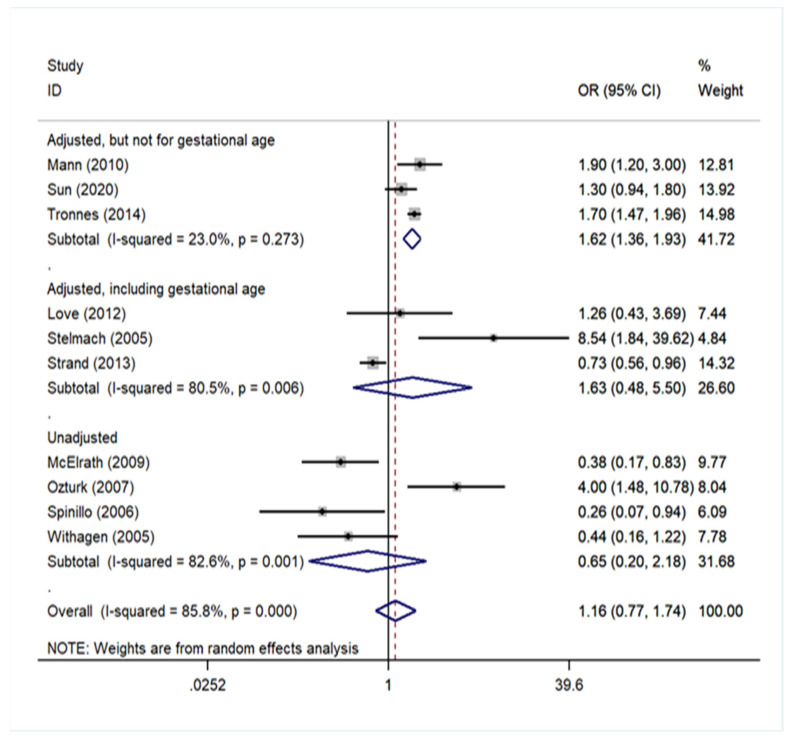
Forest plot describing the association between preeclampsia and cerebral palsy and the subgroup analysis by degree of adjustment.

**Figure 2 children-09-00385-f002:**
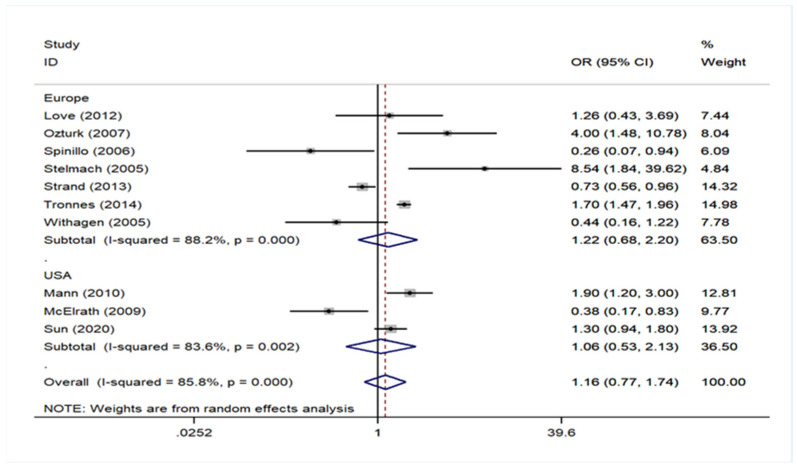
Forrest plot describing the association between preeclampsia and cerebral palsy based on the subgroup analysis of studies depending on geographical region.

**Figure 3 children-09-00385-f003:**
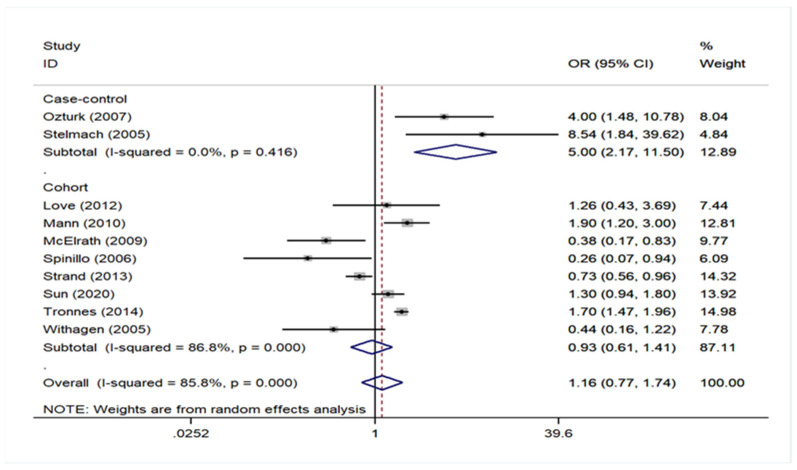
Forest plot describing the association between preeclampsia and cerebral palsy based on the subgroup analysis of studies depending on study design.

**Figure 4 children-09-00385-f004:**
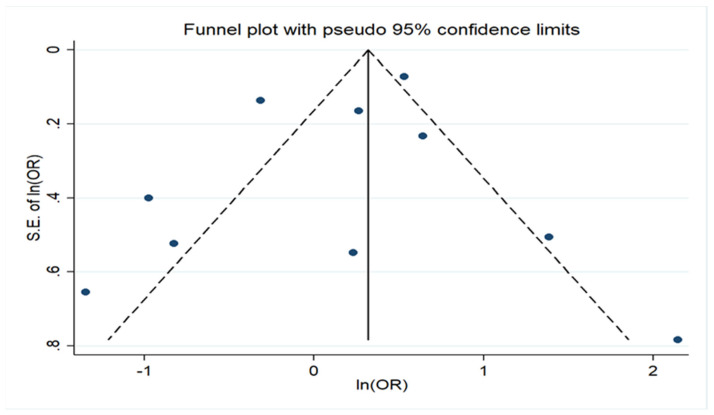
Funnel plot of the meta-analysis on association of preeclampsia with cerebral palsy showing evidence of publication bias.

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
