# Peer review of "Preeclampsia and Cerebral Palsy in Offspring"

_children, 2022, doi:10.3390/children9030385_

Round 1
Reviewer 1 Report
The manuscript is very well prepared and clearly presented however in my opinion only studies which analyzed the results adjusted for gestational age at delivey are propriate because it is most important detreminant for the outocme in relation to cerebral palsy . The second thing , I do not think one may compare early -onset with late -onset preeclampsia because for the same reason mentioned above because these are completely different subjects. (One of the study you included by Stelmach showed big differences and it what adjusted for gesttional age? )I do also recommend only to analyze early onset preeclampsia group or early- and late - onset but as seperate groups, in both cases with adjustement for gestational age at delivery
Author Response
Dear Reviewer,
Thank you very much for your time spent for the evaluation of this Systematic Review & Meta-analysis.
We considered gestational age as a continuous variable because it has been proposed that early and late-PE are not two different clinical entities, but rather a spectrum of the same disorder, the degree of which is reflected in both the gestational age at the time of delivery and the severity of the disease based on clinical and laboratory findings, according to Akolekar et al. 2013 (Akolekar R, Syngelaki A, Poon L, Wright D, Nicolaides KH. Competing Risks Model in Early Screening for Preeclampsia by Biophysical and Biochemical Markers. Fetal Diagn Ther. 2013;33(1):8–15).
I am sending you the revised manuscript as well.
Kind Regards,
Dr. Miltiadis Badagionis

Reviewer 2 Report
This systematic review examines the relationship between preeclampsia and cerebral palsy. In the meta-analysis, the authors found that pre-eclampsia is not associated with cerebral palsy. I have several concerns about this manuscript and have suggestions to improve the manuscript. My comments are listed below.
Lines 39-44
Suggest adding pre-gestational diabetes and twins as risk factors of pre-eclampsia.
Line 85
Please cite the latest version of PRISMA guidelines.
Line 85
A meta-analysis requires registration in PROSPERO. Have the authors registered?
Please clarify the keywords for the literature search. Please use MeSH keywords for the PubMed search.
Line 139
I cannot see Figure 1, Table 1, Table 2, and Table 4.
Figure 2
Please check the data carefully. For instance, Sacks et al. have shown that pre-eclampsia was associated with an increased rate of neuropsychiatric hospitalizations (adjusted OR 1.36 95%CI 1.14-1.64). I think the data about cerebral palsy is not available in this study.
The definition of pre-eclampsia may be different among the studies. What did the authors think about this problem?
Author Response
Dear Reviewer,
Thank you very much for your time spent for the evaluation of this Systematic Review & Meta-analysis.
We have made all the changes you suggested to us!!
We have also made a registration to PROSPERO and we are waiting for the response.
We must tell you that all the tables and figures you asked for (Figure 1, Table 1, Table 2, and Table 4) are available to the Supplemental Tables and Figures of this article that we have already sent, but we are sending the revised Supplemental Figures and Tables with the revised manuscript.
Also, thank you very much for the comment on one of the studies that did not provide the right odds ratio regarding our analysis. We took care to remove this study as irrelevant for our analysis and we came up with new results without affecting our conclusions.
Definition of preeclampsia may be different among the studies and that is something that we cannot avoid given that the studies cover different decades and different populations. Also, the number of studies on the association that we examine are very limited and we tried to include all available data in our analysis.
Kind Regards,
Dr. Miltiadis Badagionis

Round 2
Reviewer 1 Report
I do agree with Your explanation. I think that discussion is very well written pointing out the possible crucial role for gestational age at delivery. I think it may be suffcient for publish although I do recommend some little improvements:
- introduction is much too long concerning the description of preeclampsia , including patophysiology, is not necessery, I recommend to make it shorter with some information on the possible link between PE and CP because it concerns the topic of Your study
Author Response
Dear Reviewer,
Thank you very much for your time spent for the evaluation of this Systematic Review.
We took into consideration all your comments and we made some changes in the introduction of this manuscript, as we replaced a small part of the pathophysiology of preeclampsia with a part regarding the association between preeclampsia and cerebral palsy (bold text on the manuscript)
We are sending back the revised manuscript with all the changes, as we also added some references
Kind Regards,
Dr. Miltiadis Badagionis

Reviewer 2 Report
The authors could revise the manuscript well. Nevertheless, PROSPERO registration is required upon completion of data extraction (Guidance notes for registering a systematic review protocol with PROSPERO: https://www.crd.york.ac.uk/prospero/documents/Registering%20a%20review%20on%20PROSPERO.pdf; page 3, last paragraph). Therefore, the authors cannot register this systematic review at this phase.
Author Response
Dear Reviewer,
Thank you very much for your time spent for the evaluation of this Systematic Review.
We made some changes in the introduction of this manuscript, as we replaced a small part of the pathophysiology of preeclampsia with a part regarding the association between preeclampsia and cerebral palsy (bold text on the manuscript)
We are sending back the revised manuscript with all the changes, as we also added some references
As far as PROSPERO registration is concerned, we have already sent the protocol in order to be registered, but we do not have any response yet. As we have seen in the past, this is a process that takes enough time and in similar cases there was no problem since the protocol has been submitted. We hope that this is not a serious issue regarding the publication of this Systematic Review.
Kind Regards,
Dr. Miltiadis Badagionis
